# ChineseVideoBench: Benchmarking Multimodal Large Models for Chinese Video Question Answering

## Abstract

This paper introduces ChineseVideoBench, a pioneering benchmark specifically designed for evaluating Multimodal Large Language Models (MLLMs) in Chinese Video Question Answering. The growing demand for sophisticated video analysis capabilities highlights the critical need for comprehensive, culturally-aware evaluation frameworks. ChineseVideoBench addresses this gap by providing a robust dataset and tailored evaluation metrics, enabling rigorous assessment of state-of-the-art MLLMs on complex Chinese video content. Specifically, ChineseVideoBench comprises 8 main classes and 12 sub-classes, encompassing tasks that demand both deep video understanding and nuanced Chinese linguistic and cultural awareness. Our empirical evaluations reveal that ChineseVideoBench presents a significant challenge to current MLLMs. Among the models assessed, Gemini 2.5 Pro achieves the highest performance with an overall score of 77.9%, while InternVL-38B emerges as the most competitive open-source model.

## 1 Introduction

The rapid evolution of Large Language Models (LLMs) has fundamentally reshaped the landscape of artificial intelligence, paving the way for Multimodal Large Language Models (MLLMs) that extend their powerful reasoning capabilities to the visual domain (Awadalla et al., 2023; Li et al., 2022; Liu et al., 2023). While initial breakthroughs focused on static images, the research frontier has extended to the dynamic, information-rich medium of video. This has spurred a wave of innovation, yielding numerous video-capable MLLMs designed to interpret and reason about temporal data (Maaz et al., 2023; Zhang et al., 2023; Lin et al., 2023a).

However, the development of robust evaluation frameworks has critically lagged behind model creation. A significant majority of existing video question-answering benchmarks are overwhelmingly English-centric (Li et al., 2024c; Ning et al., 2023; Fu et al., 2025; Liu et al., 2024; Xu et al., 2016; Fang et al., 2024b). This linguistic bias creates a substantial blind spot, as it prevents a fair assessment of MLLM performance in diverse cultural and linguistic contexts. The problem is especially pronounced for the Chinese language; with the world's largest population of speakers and a colossal, thriving ecosystem of digital video content, Chinese videos are often embedded with unique cultural references, idioms, and context-specific knowledge that are opaque to models trained primarily on English-language data. Consequently, while many MLLMs claim strong multilingual and video understanding abilities, there has been no specialized, high-quality benchmark to systematically validate these claims for Chinese video content.

To address this critical research gap, we introduce ChineseVideoBench, a new, comprehensive benchmark meticulously designed for the nuanced task of Chinese Video Question Answering. ChineseVideoBench is built upon a foundation of 1,625 high-quality videos, carefully curated to span 11 distinct real-world domains such as travel, food, news, and education. To ensure a focused evaluation of visual comprehension, all audio tracks have been removed, compelling models to rely solely on visual information. The videos average approximately one minute in duration, a length chosen to balance content richness with evaluation efficiency (see Figure 5).

A rigorous, human-centric annotation process guides the construction of our benchmark to ensure unparalleled quality and relevance. As depicted in Figure 2, our data pipeline is executed entirely by

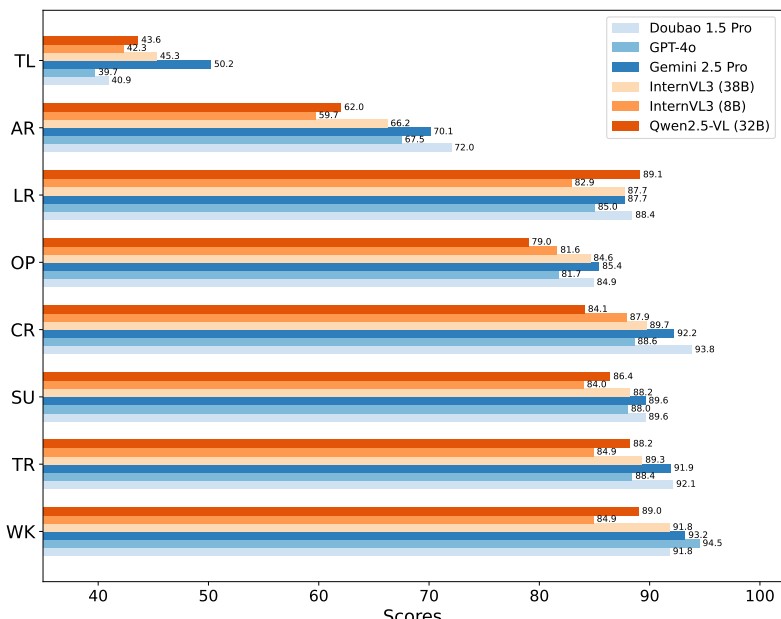

Figure 1: **Performance comparison of open-source and closed-source MLLMs on ChineseVideoBench** across eight tasks: world knowledge (WK), topic recognition (TR), scene understanding (SU), character recognition (CR), temporal localization (TL), object perception (OP), action recognition (AR), and logical reasoning (LR).

a team of nine professional native Chinese-speaking annotators. Videos are sourced from platforms with CC0-licenses, and a strict filtering protocol is enforced to exclude sensitive content. We devise a three-stage annotation workflow: (1) initial generation of question-answer pairs by one group of annotators; (2) independent review and filtering of these pairs by a second group; and (3) final validation by a third group that answer the questions based solely on the video. This process yield 6,507 high-quality multiple-choice questions, each designed to probe a specific capability. The questions are hierarchically organized into eight main task categories, including character recognition, temporal localization, and world knowledge which are further divided into twelve sub-tasks (see Figure 3). This structure allows for a granular analysis of model performance across both fine-grained detail perception and holistic content understanding.

We conduct an extensive evaluation on ChineseVideoBench, benchmarking a wide array of influential MLLMs to establish a comprehensive performance landscape. The models evaluated include leading proprietary systems such as GPT-4o (OpenAI, 2024), Gemini 2.5 Pro (Comanici et al., 2025), and Doubao 1.5 Vision Pro (Guo et al., 2025), alongside prominent open-source models like the InternVL3 (Zhu et al., 2025) series, the Qwen2.5-VL (Bai et al., 2025) series, and various LLaVA-based architectures (Lin et al., 2023a). Our findings, summarized in Tables 2 and 3, reveal that ChineseVideoBench poses a formidable challenge to the current state of the art. Critically, our evaluation reveals a distinct performance gap: **we find that models trained primarily on English-language data and benchmarks do not generalize well to our benchmark**, systematically underperforming on tasks that require deep cultural or linguistic understanding. While **Gemini 2.5 Pro** achieves the highest overall score, its performance indicates that significant challenges remain, underscoring the limitations of current paradigms.

This paper's contributions are designed not only to measure performance but also to illuminate a path forward. We believe our work can actively guide the optimization of future models. Our contributions are:

- We introduce ChineseVideoBench, the first large-scale, human-annotated benchmark for Chinese VideoQA, specifically designed to test for deep linguistic and cultural understanding.

Table 1: **Comparison of ChineseVideoBench with existing video benchmarks**. #Videos represents the number of videos in each benchmark. #QA pairs represents the number of questions in each benchmark. Anno. indicates the annotation method: human annotation (M) or automatically generated (A). Lang. indicates the language of questions and answers in each benchmark.

| Benchmarks | #Videos | #QA pairs | Anno. | Lang. |
|---|---|---|---|---|
| Video-MME | 900 | 2700 | M | English |
| MMBench-Video | 609 | 1998 | M | English |
| Video-Bench | 5917 | 17036 | A&M | English |
| MVBench | 3641 | 4000 | A | English |
| TempCompass | 410 | 7540 | A&M | English |
| ChineseVideoBench | 1625 | 6507 | M | Chinese |

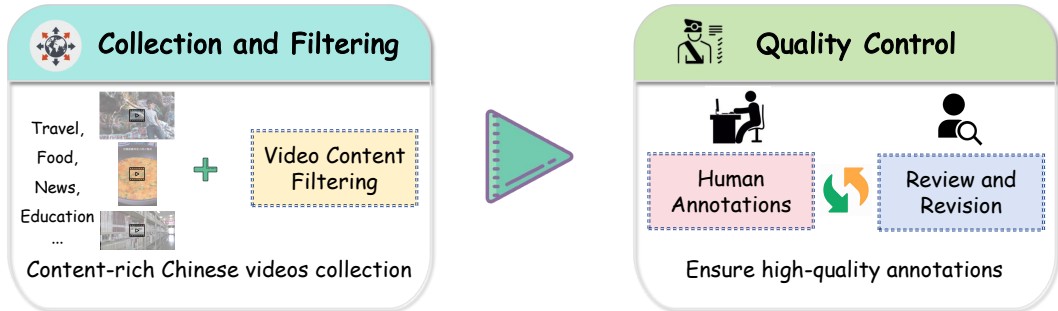

Figure 2: **Construction pipeline of ChineseVideoBench**. We employ a multi-tier annotation process conducted entirely by human annotators to construct the benchmark.

- We detail our high-quality data construction pipeline and the benchmark's hierarchical design, which enables a fine-grained diagnosis of model weaknesses.
- We provide a thorough empirical analysis of over a dozen leading MLLMs, establishing strong baselines and demonstrating that proficiency on English-centric tasks does not readily transfer to the complexities of Chinese video content.

## 2 RELATED WORK

**MLLM.** Recent breakthroughs in Large Language Models (LLMs) (Touvron et al., 2023a;b; OpenAI, 2022; Team, 2024; GLM et al., 2024; Team, 2023) have laid a solid foundation for the development of Multimodal Large Language Models (MLLMs) (Awadalla et al., 2023; Liu et al., 2023; Li et al., 2022; 2023; Wang et al., 2024a; Bai et al., 2025; Wang et al., 2024b; Zhu et al., 2025). Researchers have rapidly transferred the powerful reasoning and generative capabilities of LLMs to the visual domain, giving rise to MLLMs. Early explorations primarily focused on effectively bridging the modality gap between vision and text. Various innovative connection strategies were proposed, such as integrating multimodal features via dedicated cross-attention layers, or designing a lightweight Querying Transformer as bridges between visual encoders and language models (Awadalla et al., 2023; Li et al., 2022; Dai et al., 2023; Li et al., 2023). The LLaVA (Liu et al., 2023) framework bridges vision and language by employing a linear projection layer. Additionally, models such as Fuyu-8B (Bavishi et al., 2023) and VoRA (Wang et al., 2025) represent a paradigm shift by forgoing a dedicated vision encoder, instead adopting a novel architecture that directly integrates image features into LLMs. As research shifted from static images to dynamic videos, these architectures were extended accordingly. Some works attempted to align video frame features through simple linear projections (Maaz et al., 2023), while others designed more complex dynamic query mechanisms to capture temporal information (Zhang et al., 2023; Li et al., 2024b). To achieve a more comprehensive understanding of videos, subsequent models further explored joint training strategies on mixed image-text and video data (Lin et al., 2023a). Our work, ChineseVideoBench, will systematically evaluate these representative open-source and closed-source models (OpenAI,

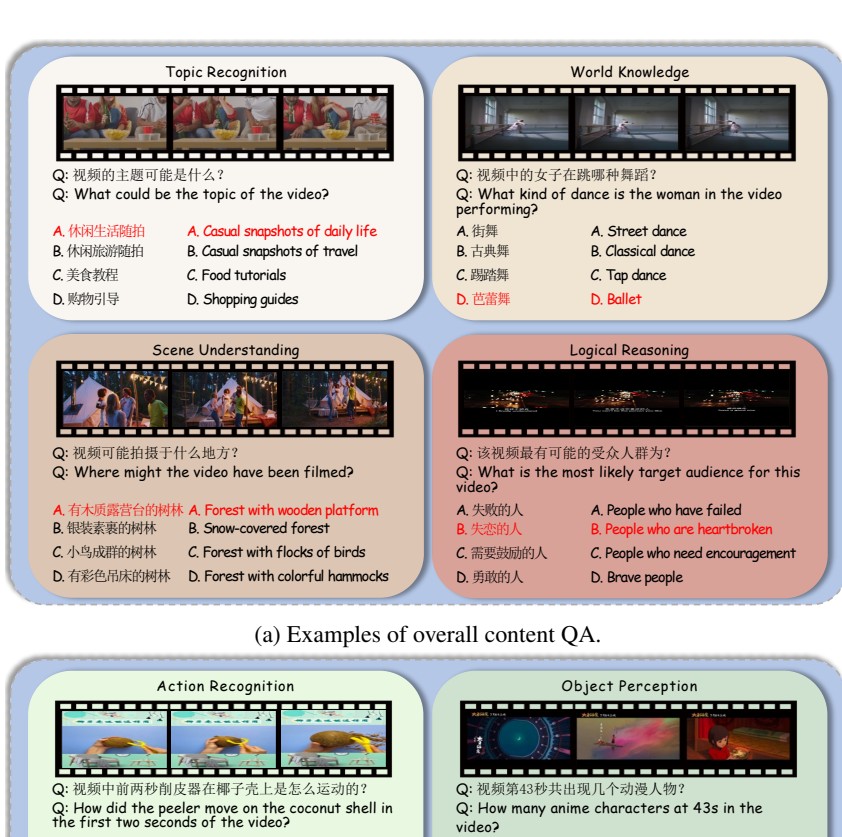

(a) Examples of overall content QA.

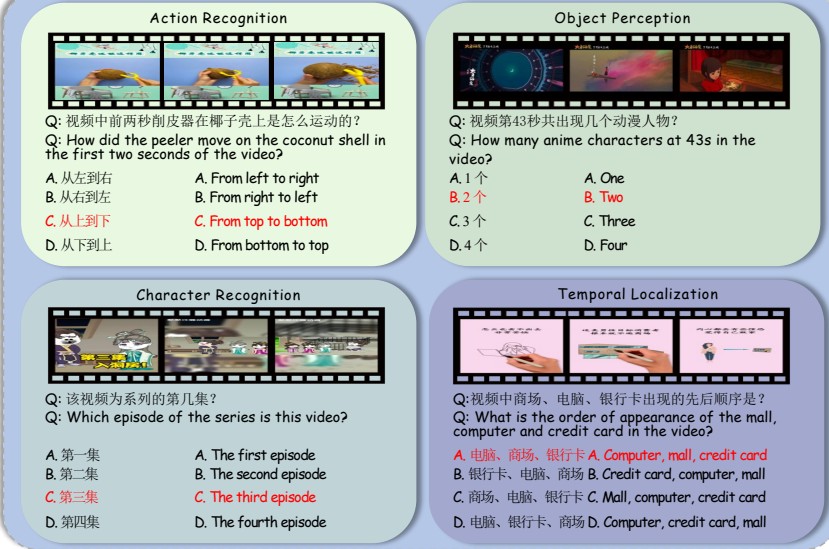

(b) Examples of detailed content QA.

Figure 3: **Representative QA examples from different tasks.** Each example displays selected video frames, Chinese QA pairs, and corresponding English translations. Correct answer options are highlighted in red.

2023; Gemini Team, 2023; Guo et al., 2025) to investigate their specific performance in processing Chinese videos.

**VideoQA Benchmarks.** Video Question Answering is a core task for measuring whether models truly "understand" videos. While existing VideoQA benchmarks (Li et al., 2024b; Fang et al., 2024a; Mun et al., 2017; Tapaswi et al., 2016) are diverse, they suffer from several fundamental limitations. Many (Xu et al., 2016; Li et al., 2024c; Liu et al., 2024) predominantly utilize short video clips, which are disconnected from the information-dense, long-form videos (e.g., documentaries, educational lectures) common in real-world scenarios, thus failing to evaluate a model's ability to capture long-range dependencies effectively.

Although some work has attempted to use LLMs as evaluators (Maaz et al., 2023), the stability and alignment of this approach with human judgment remain questionable. Most critically, existing high-quality benchmarks are almost entirely English-centric, overlooking the crucial role of linguistic and cultural contexts in deep video understanding. To systematically address these challenges, we introduce ChineseVideoBench. As the first comprehensive benchmark designed specifically for Chinese long-video scenarios, it features two key characteristics: 1) Its videos are sourced from the real-world Chinese Internet, enhancing their practical relevance; 2) it features open-ended question-answer pairs that are fully manually annotated to ensure high quality and accuracy, designed to probe the models' capabilities for in-depth reasoning and holistic understanding. ChineseVideoBench aims to fill the gap in Chinese video understanding evaluation and establish a reliable benchmark for future research.

## 3 CHINESEVIDEOBENCH

We present ChineseVideoBench, a benchmark designed to facilitate a comprehensive and fine-grained evaluation of MLLMs on Chinese video understanding. In this section, we detail its rigorous data construction process, its hierarchical task structure, and statistics.

### 3.1 DATA COLLECTION

The primary goal of ChineseVideoBench is to move beyond surface-level object recognition and evaluate a model's ability to comprehend content embedded in a specific linguistic and cultural context. To achieve this, we curate a dataset from real-world scenarios relevant to a Chinese-speaking audience. The complete data construction pipeline is shown in Figure 2.

To ensure ethical and broad usage, we manually collect videos from CC0-licensed platforms. The collection is performed by annotators, who follow strict guidelines to exclude videos containing identifiable human faces, sensitive political content, or dangerous material. We gather content from 11 diverse domains to ensure thematic variety. A key criterion for selection is the presence of rich visual content and in-video Chinese text, providing a natural testbed for OCR and other text-related tasks. After an initial collection of approximately 10,000 videos, we filter this pool down to the final 1,625 videos that possess sufficient complexity for generating challenging question-answer pairs. To isolate visual understanding capabilities and ensure broad model compatibility, all audio tracks are removed.

### 3.2 HIERARCHICAL TASK DESIGN AND ANNOTATION

To enable a fine-grained diagnosis of model capabilities, we design a hierarchical evaluation framework and populate it using a meticulous human annotation process.

**Hierarchical Task Framework.** The questions are first divided into two primary aspects: **Overall Content Comprehension** (holistic understanding) and **Detailed Content Understanding** (fine-grained perception). These are further broken down into eight task categories and twelve specialized sub-tasks, as shown in Figure 4. The categories include standard perception tasks (e.g., object and action recognition) as well as tasks specifically tailored to our focus, such as **Chinese OCR**, **Reasoning**, and **Chinese World Knowledge**. This hierarchical structure allows us to pinpoint specific model strengths and weaknesses with high precision.

**Annotation Process.** We employ a rigorous, three-stage human annotation process to ensure the highest data quality. All nine annotators are native Chinese speakers with extensive experience and cultural knowledge.

1. **Generation:** The first group of annotators watches each video and creates approximately six multiple-choice questions per video, distributed across the predefined task categories. Each question included a correct answer and three plausible, challenging distractors designed to test for deep understanding.

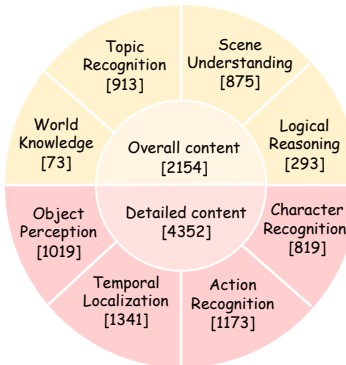 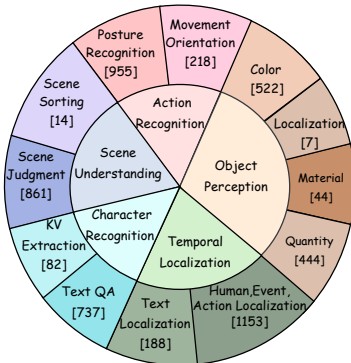

Figure 4: **Question distribution of ChineseVideoBench**. **Left**: Question distribution of main aspects and tasks; **Right**: Question distribution of sub-tasks; Numbers in square brackets indicate the number of questions for each aspect, task, and sub-task.

2. **Verification:** A second, independent group reviews each video and its associated Q&A pairs. They correct ambiguities, improve the quality of distractors, and discard any low-quality or flawed items.

3. **Validation:** A third group answered the questions after watching the videos, serving as a final human performance check and ensuring the questions are solvable, clear, and unambiguous.

This multi-tier, fully manual process eliminates the noise and potential biases of automated generation methods and ensures that our benchmark is both challenging and fair. The final result is a high-quality dataset of 6,507 validated question-answer pairs. We opt for a multiple-choice format over open-ended questions to allow objective, stable, and automated evaluation, avoiding the inconsistencies of LLM-based judges (OpenAI, 2024).

## 3.3 DATASET STATISTICS

The final ChineseVideoBench benchmark is a large-scale, high-quality resource designed for robust MLLM evaluation. In this section, we provide a detailed statistical breakdown of its composition.

**Video Distribution.** The benchmark is composed of **1,625** unique videos. These videos are intentionally diverse, sourced from **11 distinct domains** to ensure comprehensive coverage of real-world content. These domains include Food, Sports, Education, Entertainment, Travel, News, Film, Music, Technology, Dance, and Gaming, reflecting a wide spectrum of common video genres. In terms of duration, the average video length is approximately one minute, with the majority of videos falling under the five-minute mark. The precise distribution of video durations is illustrated in the top panel of Figure 5. This duration profile ensures a rich temporal context for tasks like action recognition and temporal reasoning.

**Question-Answering Pairs.** The core of our benchmark is its **6,507** meticulously curated question-answer pairs. Each entry is a multiple-choice question with a single correct answer and three carefully designed, plausible distractors. The questions are organized into our hierarchical structure for fine-grained analysis:

- **Two Primary Aspects:** There are **4,352** questions targeting **Detailed Content Understanding** (e.g., identifying specific objects or text) and **2,154** questions targeting **Overall Content Comprehension** (e.g., summarizing the topic or inferring intent).

- **Task Categories:** These aspects are further broken down into **eight main task categories** and **twelve specialized sub-tasks**. The distribution of questions across these categories is visualized in the left panel of Figure 4, showing a balanced representation of both fundamental perception tasks and advanced reasoning challenges.

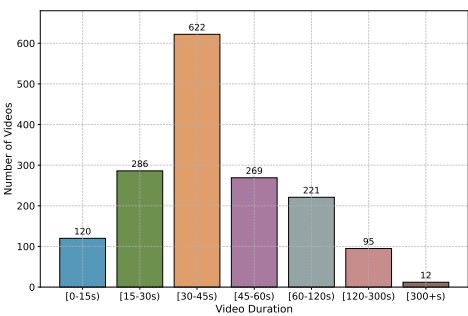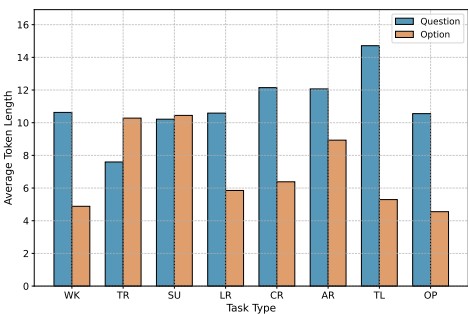

Figure 5: **Left**: Distribution of video durations; **Right**: Distribution of token lengths for questions and options across tasks, tokenized using GPT4o OpenAI (2024). The tasks include world knowledge (WK), topic recognition (TR), scene understanding (SU), character recognition (CR), temporal localization (TL), object perception (OP), action recognition (AR), and logical reasoning (LR).

Table 2: **Evaluation results of MLLMs on ChineseVideoBench.** Experimental results are evaluated across two main domains (overall content and detailed content) and eight distinct tasks: world knowledge (WK), topic recognition (TR), scene understanding (SU), character recognition (CR), temporal localization (TL), object perception (OP), action recognition (AR), and logical reasoning (LR). The **Overall** refers to the overall accuracy across the entire benchmark. Model abbreviations are defined as follows: Doubao 1.5 VP (Doubao 1.5 Vision Pro), LLaVA-OV (LLaVA-One-Vision), VideoChat2-M (VideoChat2-Mistral), and Chat-UniVi-V (Chat-UniVi-V1.5).

| Model | LLM Param | Overall (%) | Domain(%) | | Task(%) | | | | | | | |
|---|---|---|---|---|---|---|---|---|---|---|---|---|
| | | | Overall Content | Detailed Content | WK | TR | SU | CR | TL | OP | AR | LR |
| Human | - | 94.8 | 93.5 | 95.5 | 97.3 | 94.0 | 92.9 | 97.3 | 95.3 | 95.2 | 94.7 | 92.8 |
| Closed-source MLLMs | | | | | | | | | | | | |
| Doubao 1.5 VP | - | 76.5 | **90.6** | 69.5 | 91.8 | **92.1** | **89.6** | **93.8** | 40.9 | 84.9 | **72.0** | **88.4** |
| GPT4o | - | 73.4 | 88.0 | 66.2 | **94.5** | 88.4 | 88.0 | 88.6 | 39.7 | 81.7 | 67.5 | 85.0 |
| o4-mini | - | 74.2 | 87.4 | 67.7 | 91.8 | 87.1 | 87.5 | 89.7 | 43.8 | 82.7 | 66.4 | 87.0 |
| Gemini 2.5 Pro | - | **77.9** | 90.4 | **71.7** | 93.2 | 91.9 | **89.6** | 92.2 | **50.2** | **85.4** | 70.1 | 87.7 |
| Open-source MLLMs | | | | | | | | | | | | |
| LLaVA-OV | 7B | 62.3 | 76.4 | 55.2 | 83.6 | 72.8 | 78.3 | 54.7 | 38.4 | 76.3 | 56.6 | 79.9 |
| VideoChat2-M | 7B | 32.6 | 32.3 | 32.8 | 27.4 | 30.7 | 36.6 | 20.9 | 17.7 | 46.0 | 46.8 | 25.6 |
| Chat-UniVi-V | 7B | 37.9 | 48.8 | 32.5 | 49.3 | 44.0 | 47.2 | 29.8 | 18.3 | 40.0 | 44.2 | 67.6 |
| LLaVA-Video | 7B | 64.8 | 78.9 | 57.9 | 84.9 | 74.7 | 83.0 | 61.1 | 40.0 | 80.0 | 56.9 | 78.2 |
| Video-LLaVA | 7B | 32.4 | 37.1 | 30.1 | 39.7 | 30.9 | 40.5 | 29.4 | 16.1 | 35.5 | 41.9 | 45.7 |
| VILA-1.5 | 8B | 45.4 | 57.1 | 39.7 | 74.0 | 53.7 | 59.1 | 33.7 | 31.4 | 60.1 | 35.5 | 57.7 |
| NVILA | 8B | 61.7 | 78.0 | 53.7 | 86.3 | 73.8 | 81.0 | 58.4 | 32.7 | 75.1 | 55.8 | 79.5 |
| Qwen2-VL | 7B | 68.0 | 85.0 | 59.7 | 93.2 | 85.4 | 84.3 | 83.0 | 36.8 | 76.0 | 55.3 | 83.3 |
| Qwen2.5-VL | 7B | 69.8 | 85.4 | 62.1 | 89.0 | 86.3 | 84.5 | 82.7 | 42.1 | 76.3 | 58.1 | 84.3 |
| Qwen2.5-VL | 32B | 72.1 | 87.6 | 64.5 | 89.0 | 88.2 | 86.4 | 84.1 | 43.6 | 79.0 | 62.0 | 89.1 |
| InternVL3 | 8B | 71.2 | 84.3 | 64.8 | 84.9 | 84.9 | 84.0 | 87.9 | 42.3 | 81.6 | 59.7 | 82.9 |
| InternVL3 | 38B | 75.2 | 88.7 | 68.5 | 91.8 | 89.3 | 88.2 | 89.7 | 45.3 | 84.6 | 66.2 | 87.7 |

**Linguistic Properties.** To ensure fairness and consistency, we analyzed the linguistic properties of the questions and options. As illustrated in the bottom panel of Figure 5, we calculated the average token length for both questions and their corresponding options across all task categories. The results demonstrate a remarkably stable and consistent length profile. This uniformity ensures that no single task is inherently more difficult due to greater linguistic complexity, allowing for a more direct and fair comparison of a model's underlying video understanding capabilities across different tasks and establishing a stable foundation for reliable benchmarking.

Table 3: **Evaluation results of MLLMs on various sub-tasks.** Abbreviations used include: Doubao 1.5 VP (Doubao 1.5 Vision Pro), LLaVA-OV (LLaVA-One-Vision), VideoChat2-M (VideoChat2-Mistral), KV (key-value extraction), TQA (text question answering), Loc. (localization), Mat. (material), Quant. (quantity), SS (scene sorting), SJ (scene judgment), MO (movement orientation), PR (posture recognition), and H./E./A. (Human/Event/Action localization).

| Models | LLM Param | CR(%) | | OP(%) | | | | SU(%) | | AR(%) | | TL(%) | |
|---|---|---|---|---|---|---|---|---|---|---|---|---|---|
| | | KV | TQA | Loc. | Color | Mat. | Quant. | SS | SJ | MO | PR | H./E./A. | Text |
| Human | - | 95.1 | 97.6 | 100 | 96.0 | 100 | 93.7 | 92.9 | 92.9 | 91.7 | 95.4 | 95.0 | 97.3 |
| Closed-source MLLMs | | | | | | | | | | | | | |
| Doubao 1.5 VP | - | **95.1** | **93.6** | **100** | **93.7** | 93.2 | 73.4 | **85.7** | **89.7** | 43.6 | **78.4** | 41.6 | 36.7 |
| GPT4o | - | 90.2 | 88.5 | 85.7 | 88.5 | 90.9 | 73.0 | 78.6 | 88.2 | 45.9 | 72.5 | 39.5 | 40.4 |
| o4-mini | - | 91.5 | 89.6 | 85.7 | 89.8 | 95.5 | 73.0 | **85.7** | 87.6 | 50.5 | 70.1 | 44.6 | 39.4 |
| Gemini 2.5 Pro | - | 92.7 | 92.1 | 85.7 | 93.1 | **97.7** | **75.0** | **85.7** | **89.7** | **52.3** | 74.1 | **51.0** | 45.2 |
| Open-source MLLMs | | | | | | | | | | | | | |
| LLaVA-OV | 7B | 58.5 | 54.3 | 71.4 | 81.8 | 88.6 | 68.5 | 57.1 | 78.6 | 36.2 | 61.3 | 39.5 | 31.9 |
| VideoChat2-M | 7B | 18.3 | 21.2 | 28.6 | 55.0 | 27.3 | 37.6 | 28.6 | 36.8 | 39.9 | 48.4 | 18.6 | 12.8 |
| Chat-UniVi-V1.5 | 7B | 23.2 | 30.5 | 28.6 | 48.5 | 25.0 | 31.8 | 28.6 | 47.5 | 39.4 | 45.3 | 19.2 | 12.8 |
| LLaVA-Video | 7B | 59.8 | 61.2 | 71.4 | 87.2 | 86.4 | 70.9 | 71.4 | 83.2 | 40.4 | 60.6 | 40.7 | 35.6 |
| Video-LLaVA | 7B | 24.4 | 30.0 | 71.4 | 36.8 | 34.1 | 34.0 | 28.6 | 40.7 | 44.0 | 41.4 | 17.0 | 10.6 |
| VILA-1.5 | 8B | 41.5 | 32.8 | 28.6 | 67.4 | 38.6 | 53.8 | 57.1 | 59.1 | 23.4 | 38.3 | 31.6 | 30.3 |
| NVILA | 8B | 57.3 | 58.5 | 85.7 | 82.8 | 90.9 | 64.2 | 71.4 | 81.2 | 35.3 | 60.4 | 32.4 | 35.1 |
| Qwen2-VL | 7B | 89.0 | 82.4 | 71.4 | 84.7 | 86.4 | 64.6 | 78.6 | 84.4 | 37.2 | 59.5 | 36.5 | 38.3 |
| Qwen2.5-VL | 7B | 87.8 | 82.1 | 57.1 | 83.5 | 90.9 | 66.7 | 78.6 | 84.6 | 39.4 | 62.3 | 41.8 | 44.1 |
| Qwen2.5-VL | 32B | 89.0 | 83.6 | **100** | 86.8 | 90.9 | 68.2 | 71.4 | 86.6 | 42.7 | 66.4 | 42.5 | 50.5 |
| InternVL3 | 8B | 90.2 | 87.7 | 85.7 | 89.8 | 86.4 | 71.4 | 92.9 | 83.9 | 41.3 | 63.9 | 42.4 | 41.5 |
| InternVL3 | 38B | 91.5 | 89.6 | **100** | 92.1 | 90.9 | 74.8 | **85.7** | 88.3 | 46.3 | 70.8 | 45.0 | **47.3** |

# 4 EXPERIMENTS

This section presents experiments conducted on various open-source and commercial closed-source models to evaluate MLLM performance on our benchmark. We first introduce the evaluation settings and then analyze model performance across different tasks.

## 4.1 EVALUATION SETTINGS

We evaluate multiple open-source and closed-source models using their default parameters and system prompts. Open-source models include Qwen2-VL-7B (Wang et al., 2024c), the Qwen2.5-VL series (7B and 32B) (Bai et al., 2025), the InternVL3 series (8B and 38B) (Zhu et al., 2025), LLaVA-One-Vision (Li et al., 2024a), VideoChat2-Mistral (Li et al., 2024b), Chat-UniVi-V1.5 (Jin et al., 2023), LLaVA-Video (Zhang et al., 2024), Video-LLaVA (Lin et al., 2023a), VILA-1.5 (Lin et al., 2023b), and NVILA (Liu et al., 2025). Closed-source models comprise GPT-4o (OpenAI, 2024), o4-mini (OpenAI, 2025), Gemini 2.5 Pro (Comanici et al., 2025), and Doubao 1.5 Vision Pro (Guo et al., 2025). The benchmark employs multiple-choice questions with fixed prompts that input video frames, corresponding questions, and options to obtain model responses. We conduct automated evaluation using accuracy as the primary metric, calculated as the proportion of correctly answered questions. This approach eliminates dependence on external LLMs (OpenAI, 2022) and ensures evaluation consistency and stability. The evaluation is based on VLMEvalkit (Duan et al., 2024).

## 4.2 QUANTITATIVE RESULTS

We conduct a comprehensive evaluation of various proprietary and open-source models on ChineseVideoBench, with detailed results presented in Tables 2 and 3.

**Performance of Proprietary Models** Proprietary models demonstrate strong performance, with overall accuracies ranging from 73.4% to 77.9%. Gemini 2.5 Pro achieves the highest score at 77.9%. Table 3 indicate that proprietary models excel in traditional visual perception tasks such

as object localization (Loc.) and material recognition (Mat.). However, they still face challenges in tasks requiring fine-grained spatiotemporal reasoning, such as temporal localization (TL) and movement orientation (MO).

**Performance of Open-Source Models**   Open-source models show remarkable progress. The top-performing InternVL3 (38B) achieves 75.2%, nearing the performance of proprietary models. Among models with smaller parameter counts (7B/8B), InternVL3 (8B) and Qwen2.5-VL (7B) stand out with accuracies of 71.2% and 69.8% respectively, significantly narrowing the gap with their larger counterparts. Nevertheless, a discernible gap remains between most open-source models and the top-tier proprietary models. As illustrated in Table 2, this gap is particularly pronounced in complex reasoning tasks. Furthermore, temporal localization (TL) remains a common bottleneck for all evaluated models, highlighting a key area for future improvement.

### 4.3 DISCUSSION

Our comprehensive evaluation provides several key insights into the current capabilities and limitations of MLLMs for video understanding.

**The Role of Culturally-Specific Training Data**   We first note that our benchmark is primarily composed of Chinese video question-answering pairs. Consequently, models extensively trained on Chinese multimodal corpora, such as InternVL3 (Zhu et al., 2025) and Qwen2.5-VL (Bai et al., 2025), exhibit a distinct advantage over peers like NVILA (Liu et al., 2025). This highlights that the linguistic and cultural context of training data is a critical factor for performance in specific scenarios, underscoring the need for more diverse datasets to build universally competent MLLMs.

**Common Failure Patterns in Spatiotemporal Reasoning**   Beyond data bias, our qualitative analysis reveals that certain task categories pose severe and consistent challenges for all evaluated models, pointing to fundamental architectural and representational weaknesses. As shown in Table 2 and Table 3, these failures are most pronounced in tasks requiring fine-grained spatiotemporal precision. The most significant bottleneck is temporal localization (TL). The poor performance is largely attributable to the prevalent video processing strategy of sparse frame sampling. This computationally efficient method often causes the model to miss short-duration events or the precise start/end moments of an action. More fundamentally, TL requires a sophisticated alignment between visual evidence and a continuous timeline. We observe that models, while often correctly identifying an event, tend to "hallucinate" plausible but inaccurate timestamps, indicating a weak temporal grounding capability. Furthermore, models exhibit significant weaknesses in tasks demanding nuanced, instance-level understanding within dynamic scenes. For instance, they struggle to perceive fine-grained action details like movement orientation (MO) and posture (PR), as subtle cues are often lost to motion blur and occlusions. Similarly, their ability for precise quantity perception (Quant.) is compromised by poor instance tracking and scene clutter. These issues highlight a broader failure in detailed spatiotemporal grounding, which is crucial for localizing text and events. In conclusion, our benchmark highlights a critical gap: while MLLMs capture the high-level gist of videos, they lack fine-grained spatiotemporal analysis, pointing to crucial future research in temporal modeling, instance tracking, and robust perception.

## 5 CONCLUSION

In this paper, we introduce ChineseVideoBench, the first large-scale, high-quality benchmark designed to address the critical lack of Chinese-centric evaluation for MLLMs in video understanding. Built upon meticulously curated videos and a rigorous human annotation pipeline, our benchmark provides a comprehensive evaluation framework for Chinese video understanding. Extensive experiments on over a dozen leading MLLMs reveal significant performance gaps and demonstrate that proficiency on English-centric benchmarks does not readily transfer to Chinese contexts. Models with stronger Chinese context, such as InternVL3, clearly outperform those trained primarily on English-language data. Despite these advantages, even top-tier models struggle with fine-grained tasks such as temporal reasoning and action recognition, highlighting common bottlenecks in the field. We believe ChineseVideoBench provides both a robust tool for measuring progress and clear, actionable insights to guide MLLM development.

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

## A APPENDIX

### A.1 IMPLEMENTATION DETAILS

This section outlines the implementation details of our experiments.

**Models.** We conduct experiments on both closed-source and open-source models. The closed-
source models evaluated are Doubao 1.5 Vision Pro (250328) (Guo et al., 2025), GPT-4o (2024-
05-13) (OpenAI, 2024), o4-mini (2025-04-16) (OpenAI, 2025), and Gemini 2.5 Pro (preview-05-
06) (Comanici et al., 2025). For the open-source models, we utilize their official checkpoints from
HuggingFace and evaluate them using their default settings and system prompts.

**Evaluation Framework.** Our evaluation pipeline is built upon the VLMEvalKit (Duan et al.,
2024). All experiments are conducted using PyTorch 2.5.1 on a single node equipped with 8
NVIDIA A100 GPUs.

**Frame Extraction.** For video analysis, we extract frames from videos as model inputs. To ensure
stable and fair comparisons, we uniformly sample 32 frames for most models, with specific sampling
strategies for certain models: 8 frames for Video-LLaVA (Lin et al., 2023a), 8 for LLaVA-One-
Vision-7B (Li et al., 2024a), 16 for VideoChat2-Mistral (Li et al., 2024c), and 1 frame per second
for Chat-UniVi-V1.5 (Jin et al., 2023).

**Evaluation Prompt.** To ensure a consistent assessment, a single instruction prompt is used for all
models. The prompt is shown in Figure 6.

```
instruction ="""

[System Prompt]

<video>

这些是视频的帧。请根据视频内容，为以下多项选择题选择最佳答案。

请仅回答正确选项的字母（A, B, C, 或 D）。

问题: {{Question}}

{{Options}}

答案:

"""
```

Figure 6: Evaluation prompt for ChineseVideoBench.

## A.2 MORE RESULTS

**Qualitative Result** Several selected examples are illustrated in Figure 7 and 8. Figure 7 involves Chinese world knowledge, where most models perform well except Video-LLaVA (Lin et al., 2023a), which provids an incorrect answer. Figrue 8 focuses on perception tasks, where both Gemini 2.5 Pro (Comanici et al., 2025) and open-source models fail, indicating ongoing challenges in specific task domains.

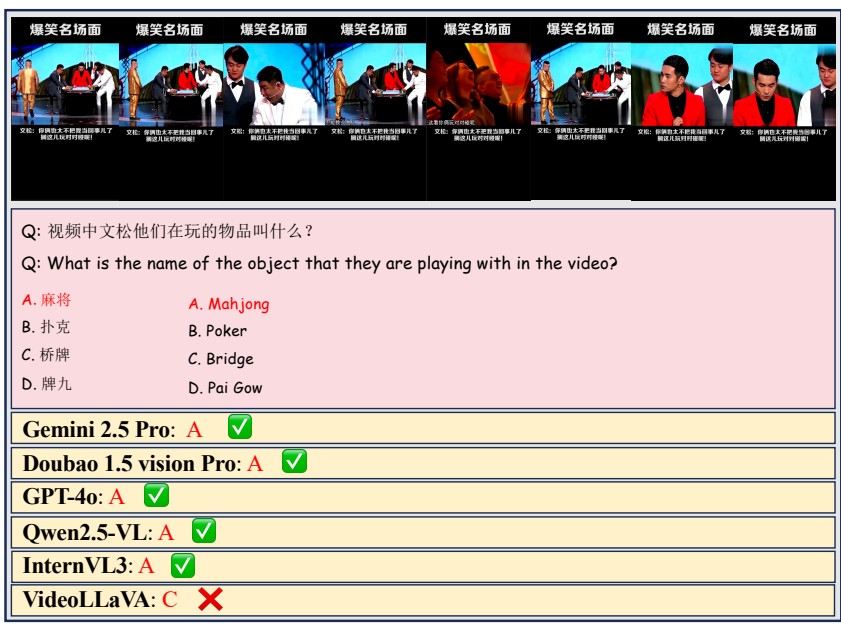

Figure 7: Response visualization of different models on ChineseVideoBench.

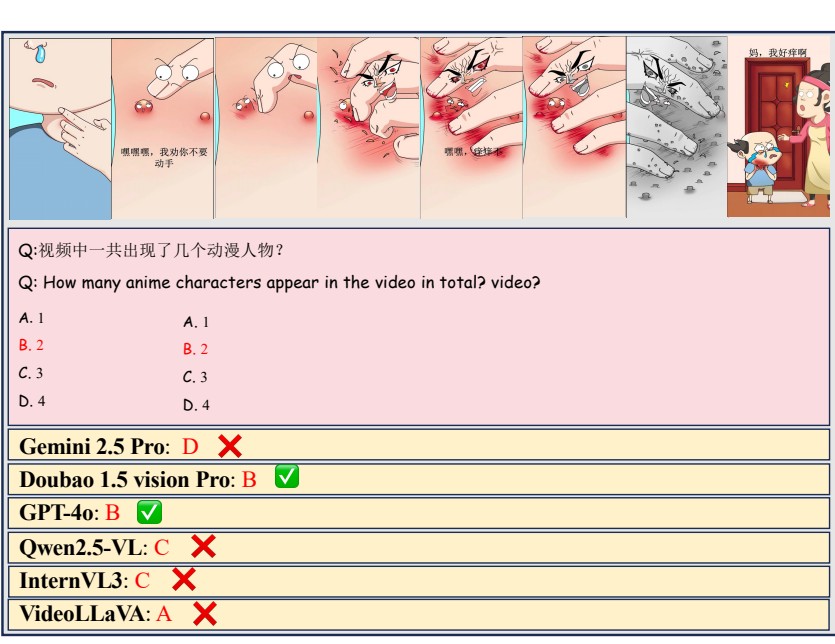

Figure 8: Response visualization of different models on ChineseVideoBench.

