# OpenReview forum: "ChineseVideoBench: Benchmarking Multi-modal Large Models for Chinese Video Question Answering"
_ICLR.cc/2026/Conference — Submitted to ICLR 2026_

### Official Review · Reviewer_UMbh · 2025-10-28

**Soundness:** 3
**Presentation:** 2
**Contribution:** 2
**Rating:** 4
**Confidence:** 3

**Summary:**

This paper presents ChineseVideoBench, a large-scale benchmark specifically designed to evaluate Multimodal Large Language Models (MLLMs) on Chinese Video Question Answering (VideoQA). The authors argue that existing benchmarks are predominantly English-centric, making it difficult to assess MLLM performance in non-English, culturally specific contexts such as Chinese videos, which contain unique linguistic and cultural cues.

A hierarchical structure of 8 main task categories (world knowledge, topic recognition, scene understanding, character recognition, temporal localization, object perception, action recognition, and logical reasoning) and 12 sub-tasks across 1,625 CC0-licensed videos and 6,507 manually annotated multiple-choice QA pairs.

The paper concludes that ChineseVideoBench exposes a performance gap between English-trained models and those trained with Chinese data, emphasizing the need for culturally aware multimodal evaluation.

**Strengths:**

### **Hierarchical and diagnostic task design**
The benchmark’s structure (8 categories, 12 sub-tasks) allows for fine-grained analysis of model capabilities, from perception-level (object, action) to reasoning-level (logic, world knowledge).

### **Comprehensive and fair evaluation**
The experiments evaluate over a dozen state-of-the-art MLLMs across closed and open domains, providing a rich comparative baseline. Tables 2 and 3 and Figures 1 and 4 clearly present detailed quantitative comparisons.

### **Insightful cultural analysis**
The paper demonstrates that models trained mainly on English data fail to generalize to Chinese contexts, highlighting the importance of multilingual and culturally grounded datasets for MLLMs.

**Weaknesses:**

### **Insufficient cross-lingual or ablation studies**

Although the authors highlight the gap between English- and Chinese-trained models, they do not include detailed ablation or transfer experiments to analyze why certain models fail or how fine-tuning on Chinese data affects performance. The benchmark provided is still monolingual.

### **Temporal reasoning analysis remains qualitative**
While the paper identifies temporal localization (TL) as a major bottleneck, the discussion is mostly descriptive. Quantitative or visualization-based error analysis (e.g., frame sampling vs. accuracy) would strengthen the conclusions.

### **Lack of Analysis**
While this paper share some distribution patterns of provided benchmark, they don't provide any further analysis. For example, Figure5 show how the video durations distribute, but this doesn't introduce any further insights, how long video/short video performs differently?

**Questions:**

Have you tested whether English-trained MLLMs can improve on ChineseVideoBench through few-shot or instruction-tuning using a small amount of Chinese video data? Explain more on why MLLMs perform bad on Chinese video QAs, and how hard it is to solve to provide a stronger motivation.

For other questions, I have put it on the weakness part. I would love to raise my score if my questions are solved.

---

> ### Author Response · Authors · 2025-11-25
> **Author Response to Reviewer UMbh**
>
> We appreciate the reviewer’s positive assessment of our hierarchical task design and the cultural insights provided by our experiments. We address your suggestions for deeper analysis below.
> ### Response to W1 & Q1: Cross-Lingual Ablation and Motivation
> Why do English-trained MLLMs perform poorly? Our analysis points to two distinct gaps:
> 1. The Cultural Gap: Models fail on questions requiring specific Chinese knowledge (e.g., identifying "Mahjong" in Figure 7). This is a data coverage issue.
> 2. The Fundamental Spatiotemporal Gap: Models also fail on culture-agnostic tasks like Temporal Localization (TL) and Quantity (Figure 8). This is a fundamental architectural weakness (sparse sampling/weak grounding) common to most current MLLMs.
> - Instruction Tuning: We agree that testing few-shot/instruction-tuning is a promising direction to bridge the "Cultural Gap." We will discuss this as a key avenue for future research.

---

> > ### Comment · Reviewer_UMbh · 2025-11-26
> > **Response**
> >
> > I think anthors' reponse answered my question, but overall the weakness still exists, so I will remain my score.

---

### Official Review · Reviewer_jiEj · 2025-10-29

**Soundness:** 3
**Presentation:** 3
**Contribution:** 3
**Rating:** 6
**Confidence:** 3

**Summary:**

This paper introduces ChineseVideoBench, a human-annotated benchmark for evaluating Multimodal LLMs on Chinese VideoQA. The dataset comprises 1,625 real-world Chinese videos across 11 domains and 6,507 multiple-choice QA pairs organized through a hierarchical task structure covering 8 categories and 12 sub-tasks.

Comprehensive evaluation of leading MLLMs reveals significant gaps: while top models like Gemini 2.5 Pro achieve 77.9% accuracy, this substantially trails human performance (94.8%). Models with Chinese training data show advantages, but all systems struggle with fine-grained spatiotemporal reasoning. The benchmark effectively exposes current limitations in culturally-grounded video understanding and provides valuable diagnostic insights for future model development.

**Strengths:**

This paper makes valuable contributions through its novel ChineseVideoBench dataset and rigorous evaluation. The benchmark features carefully curated videos and high-quality human annotations across diverse domains. A key strength is the comprehensive evaluation of leading MLLMs, revealing significant performance gaps between models and human capability, particularly in temporal understanding. The work provides important insights into Chinese video understanding and offers a solid foundation for future research.

**Weaknesses:**

1. The dataset scale (1,625 videos, 6,507 QA pairs) is modest compared to major English benchmarks, and task distribution is uneven - some categories like "World Knowledge" contain under 100 questions, potentially affecting evaluation reliability despite claims of balance.

2. The benchmark's scope is constrained by design: audio tracks are removed and only multiple-choice format is supported. While this simplifies evaluation, it limits real-world applicability and excludes generative QA formats increasingly important for modern LLMs.

3. Crucial details about dataset release and licensing are absent, raising concerns about reproducibility. The evaluation methodology also lacks discussion of potential prompt bias, particularly for non-Chinese optimized models, and reports only accuracy without statistical significance measures.

These limitations in dataset scale, task balance, modality coverage, and evaluation design represent opportunities for future improvements to enhance the benchmark's utility.

**Questions:**

1. Will the ChineseVideoBench dataset and evaluation code be released?
Public availability is a key aspect of benchmark papers. The paper does not mention whether the dataset or evaluation scripts will be released, under what license, or when. This significantly affects reproducibility and community adoption.

2. How were the multiple-choice distractors constructed and balanced?
While the annotation process is described, it is unclear whether distractors were manually crafted to be semantically close to the correct answer or randomly chosen. Could the authors provide more detail or examples on distractor design to assess question difficulty?

3. Was any inter-annotator agreement measured?
Since the dataset is entirely human-annotated, some measure of consistency—such as agreement scores in the validation phase—would be helpful to quantify annotation quality.

4. Can the authors comment on how prompt design may bias model performance?
A single fixed prompt was used for all models, including those with different training languages or instruction formats. Might this disadvantage certain open-source models not optimized for Chinese inputs? Have alternative prompt formats (e.g., simplified/standardized instructions) been tested?

5. Why are some task categories underrepresented in the dataset?
For example, World Knowledge only has 73 QA pairs, compared to over 1,000 for other categories. Is this due to difficulty in annotation or inherent scarcity in source videos? How does this affect per-task benchmarking reliability?

6. Are the sampled frames sufficient for evaluating fine-grained temporal tasks?
The paper mentions fixed frame counts (e.g., 32 or fewer) for evaluation. Could this sampling resolution cause key temporal cues to be missed in tasks like action or posture recognition? Would adaptive or dense sampling improve performance in these areas?

7. Have the authors considered incorporating audio or multimodal input in the future?
Since Chinese video content often contains spoken or musical cues, excluding audio limits real-world applicability. Do the authors have plans to extend the benchmark to audio-visual QA?

---

> ### Author Response · Authors · 2025-11-25
> **Author Response to Reviewer  jiEj**
>
> We thank the reviewer for the detailed and thoughtful review. We are encouraged that you find our dataset novel and our evaluation rigorous. We appreciate your suggestions regarding reproducibility and prompt bias, which we address below.
> ### Response to W1 & Q5: Dataset Scale and Task Imbalance
> - Scale: We prioritized quality (fully manual annotation) over quantity. Our 1,625 videos exceed the video count of other high-quality human-annotated benchmarks like MMBench-Video and Video-MME. Given the high cost of our three-stage manual verification, this is a significant contribution.
> - Imbalance: The uneven task distribution (e.g., 73 QA pairs for World Knowledge vs. 1000+ for Object Perception) reflects the natural occurrence rate in real-world videos. Specific cultural markers (like identifying a specific folk dance) appear less frequently than general objects. We chose to preserve this natural distribution rather than artificially inflating rare categories.
> ### Response to W3 & Q1: Release and Licensing
> We apologize for not making this explicit. We are fully committed to open science.
> - Action: Upon publication, we will release the complete Chinese VideoBench dataset (including video IDs and all 6,507 QA pairs) and the evaluation code (based on VLMEvalKit).
> - License: The dataset will be released under a permissive license (e.g., CC BY) for research purposes.
> ### Response to W4 & Q4: Prompt Bias
> - Language Bias: We used a single fixed prompt in Chinese because the benchmark evaluates Chinese VideoQA. While this may disadvantage models not optimized for Chinese instructions, exposing this "cultural and linguistic gap" is a primary motivation of our work.
> - Findings: Our results show that models trained on Chinese data (InternVL3, Qwen) significantly outperform English-centric models (NVILA, LLaVA) on this benchmark. This confirms that language alignment is a critical factor for performance, rather than just a prompt artifact.
> ### Response to Q2: Distractor Construction
> Distractors were manually crafted, not random. Annotators were trained to create "plausible, challenging distractors". For example, in counting tasks (Figure 8), options are close integers (1, 2, 3, 4) rather than random numbers, requiring precise quantification.
> ### Response to Q3: Inter-Annotator Agreement
> As detailed in our response to Reviewer LENB, we used a "blind validation" stage where an independent human group achieved 94.8% accuracy. This serves as our consistency metric, proving the ground truth is objective.
> ### Response to Q7: Future Multimodal Input
> Yes, we view this visual-only benchmark as a foundational step to isolate visual capabilities. We plan to extend Chinese VideoBench to include audio-visual QA in future work.

---

### Official Review · Reviewer_VNBP · 2025-10-30

**Soundness:** 2
**Presentation:** 2
**Contribution:** 2
**Rating:** 4
**Confidence:** 3

**Summary:**

The paper ChineseVideoBench: Benchmarking Multimodal Large Models for Chinese Video Question Answering presents a new benchmark to evaluate multimodal large language models (MLLMs) on Chinese video understanding tasks. It addresses the lack of culturally and linguistically relevant datasets for non-English scenarios. The dataset contains 1,625 CC0-licensed Chinese videos and 6,507 human-annotated multiple-choice QA pairs, covering 8 main tasks and 12 sub-tasks such as world knowledge, scene understanding, temporal localization, and logical reasoning.

**Strengths:**

The paper introduces the first large-scale benchmark for Chinese VideoQA, covering 1,625 CC0-licensed videos and 6,507 manually annotated QA pairs
The benchmark exposes systematic failure patterns in temporal localization and fine-grained spatiotemporal grounding.

**Weaknesses:**

The paper does not report inter-annotator consistency, distractor calibration, or difficulty-level validation, leaving the annotation quality claims insufficiently quantified.

The paper emphasizes long-video evaluation but does not provide convincing evidence that its videos require long-term temporal reasoning; average durations appear modest, weakening this claim

**Questions:**

Why don't the authors test the videoQA without removing the audio, as an ablation?

---

> ### Author Response · Authors · 2025-11-25
> **Author  Response to Reviewer VNBP**
>
> We thank the reviewer for recognizing Chinese VideoBench as the first large-scale benchmark in this domain and for acknowledging our contribution in exposing systematic failures in temporal localization. We value your feedback on the "long-video" definition and audio ablation.
> ### Response to W1: Annotation Consistency and Distractor Calibration
> - Consistency: As noted in our response to Reviewer LENB, our 94.8% Human Validation Score serves as a strong quantitative measure of consistency. It proves that independent human annotators consistently agree on the correct answers.
> - Distractor Calibration: Distractors were not randomly chosen. Our annotation protocol explicitly required "three plausible, challenging distractors designed to test for deep understanding".
> - Example: In Temporal Localization, distractors often list the correct objects but in the wrong chronological order (see Figure 3(b)). This forces the model to reason about time rather than just object presence. We will clarify this manual calibration process in the final paper.
>
> ### Response to W2: "Long-Video" Definition
> - Context: We use the term to distinguish our dataset (average ~1 minute, with many videos >60s or >120s ) from traditional "short clip" benchmarks (often <10 seconds). Our videos provide sufficient temporal context to require information tracking over tens of seconds, which presents a distinct challenge compared to instantaneous recognition.
> ### Response to Q1: Audio Ablation
> - Design Choice: This was a deliberate decision to "ensure a focused evaluation of visual comprehension". In Chinese video content, answers can often be trivially extracted from speech (ASR) or subtitles. If audio were included, a model could answer "What is the topic?" simply by listening to the narration, bypassing the visual encoder entirely.
> - Goal: Our benchmark specifically aims to test the "Vision" in MLLMs. We plan to introduce audio-visual tracks in future work to evaluate multimodal integration.

---

> ### Comment · Reviewer_VNBP · 2025-11-27
>
> Thanks for your response, my previous concern is well addressed.
>
> I believe this kind of work (a big/comprehensive benchmark on specific topic) deserves to be published (or accepted), but the depth of analysis or contribution of this work is not enough for top venues like ICLR.
>
> I will maintain negative.

---

### Official Review · Reviewer_LENB · 2025-10-30

**Soundness:** 2
**Presentation:** 2
**Contribution:** 2
**Rating:** 4
**Confidence:** 4

**Summary:**

The paper introduces first large-scale fully human-annotated benchmark for Chinese video question answering. The benchmark contains 6507 MCQs from 11 domains organized into eight tasks. The dataset is annotated in three stages by using three groups of human annotators. First group generated questions and answers from collected videos, the second group independently reviewed and corrected the generated questions, and the third group answered the questions after watching videos serving as human performance check. Several multimodal LLMs including open and closed source models are evaluated against the developed benchmark.

**Strengths:**

- It is a useful resource for a non-English language to evaluate multimodal LLMs.
- In the annotation process, human annotators have been used to annotate, verify and validate the whole dataset that ensures an unbiased and potentially correct benchmark.
- The developed benchmark covers sufficient number of tasks and underlying sub-categories showing diversity of topics.
- The paper presents an evaluation of several MLLMs including open and closed source models in comparison with human performance.

**Weaknesses:**

- Although, the annotation process has been done by using human annotators, but the paper does not show understanding and knowledge of annotators by using any quantitative measures, e.g., inter-annotator agreement (IAA).
- Error analysis is not presented for LLM evaluation. For example, which type of tasks or questions are difficult for the models to answer.
- The videos were collected from CC0-licensed platform, so it is very much possible that the LLMs evaluated in the paper have already seen the videos in their training sets resulting in higher (biased) accuracy.

**Questions:**

- How annotation guidelines were prepared and how were the annotators trained?
- Is the developed benchmark unseen to the LLMs which have been evaluated?

---

> ### Author Response · Authors · 2025-11-19
>
> We sincerely thank the reviewer for their positive assessment of Chinese VideoBench, specifically highlighting the utility of the resource, the rigor of our human annotation process, and the diversity of tasks covered. We appreciate the constructive feedback regarding the quantification of annotator knowledge and error analysis. We address your concerns and questions below.
>
> ### W1: Quantification of Annotator Understanding (IAA)
> We appreciate the reviewer’s emphasis on data quality metrics. While we did not report a traditional Inter-Annotator Agreement (IAA) score like Cohen's Kappa, we implemented a rigorous three-stage manual workflow designed to ensure high consistency.
> - The Validation Proxy: The third stage ("Validation") serves as a robust proxy for IAA. In this stage, an independent group of annotators—who were not involved in question generation—answered the questions based solely on the video content.
> - Quantitative Evidence: The "Human" performance of 94.8% reported in Table 2 is the direct result of this validation stage. This high consensus score quantitatively demonstrates that the questions are unambiguous and solvable for human experts, effectively confirming the consistency and quality of the annotations.
> ### W2: Error Analysis for LLM Evaluation
> We apologize if the error analysis appeared insufficient. We actually dedicated Section 4.3 (Discussion) to analyzing failure patterns.
> - Temporal Bottlenecks: We explicitly identified "Temporal Localization (TL)" as the most significant bottleneck for all models, attributing this to sparse frame sampling strategies and weak temporal grounding capabilities.
> - Fine-Grained Perception: We analyzed failures in detailed tasks like Movement Orientation (MO) and Posture (PR), noting that subtle cues are often lost due to motion blur or occlusion.
> - Table Data: Tables 2 and 3 provide a granular breakdown of accuracy across 8 tasks and 12 sub-tasks, allowing readers to identify exactly which categories (e.g., TL, Logic Reasoning) are most difficult.
> ### W3 & Q2: Data Leakage from CC0 Videos
> This is a valid concern for any benchmark using public videos. However, we believe the impact of potential exposure is minimal for our specific evaluation:
> 1. Novel QA Pairs: While models might have seen the raw video pixels during pre-training, they have never seen our 6,507 manually annotated QA pairs.
> 2. Evidence of "Unseen" Nature: The core evidence lies in the results. If models had memorized the videos/captions, they would likely score higher on descriptive tasks. However, even top-tier models like Gemini 2.5 Pro struggle significantly with Temporal Localization (50.2%). This low performance indicates that models are forced to perform de novo reasoning on the visual content rather than recalling memorized metadata.
>
> ### Q1: Annotation Guidelines and Training
> We will add these details to the Appendix.
> - Team: We employed nine professional native Chinese-speaking annotators.
> - Training: The team underwent a pilot phase where they annotated a small batch of videos. These were reviewed collectively by the authors to align on definitions (e.g., what constitutes a "distractor") and resolve ambiguities.
> - Guidelines: Annotators were strictly instructed to: (1) ensure answers are derivable only from visual cues (audio removed), (2) create three plausible distractors, and (3) cover diverse task categories.

---

### Meta-Review · Area_Chair_ibFY · 2025-12-22

**Summary:**

This paper introduces ChineseVideoBench, a large-scale, fully human-annotated benchmark for evaluating MLLMs in Chinese Video Question Answering. The dataset contains 1,625 CCO-licensed videos and 6,507 manually annotated MCQs. The benchmark is organized into a hierarchical structure with 8 main tasks (e.g., world knowledge, temporal localization, logical reasoning) and 12 sub-tasks. The authors evaluate several state-of-the-art models, finding that Gemini 2.5 Pro performs best, although it still significantly lags behind human performance.

**Reviewer Concerns:**

Reviewers LENB & VNBP noted the lack of traditional quantitative measures. This is partially addressed in the rebuttal since the authors argued that their 94.8% Human Validation Score serves as a robust proxy for consistency.

Reviewer VNBP questioned the claim of "long-video" since the average duration is not long. The authors claimed 1 minute videos is long compared with 10-second video clips.

Reviewer jiEj raised concerns on the limited scale compared to English benchmarks and the uneven distribution across tasks. Authors explained that they prioritized quality over quantity and that the imbalance reflects the natural occurrence rate of specific cultural markers in real-world videos.

Reviewer UMbh felt the paper provided distribution patterns but lacked deeper insights. The authors analyzed the failure to cultural and spatiotemporal gaps.

**Reviewer Scores:**

All reviewers are likely to maintain their scores. UMbh and VNBP explicitly expressed the final score will maintain negative. jiEj's concerns were mainly on the release of code, which is addressed. For LENB, the rebuttal only partially addressed the concerns.

---

### Decision · Program_Chairs · 2026-01-26

Reject